# Preliminary Report on Diet Estimation of Taruka (*Hippocamelus antisensis* d’Orbigny, 1834) in an Agricultural Area of the Andean Foothills of the Tarapacá Region, Chile

**DOI:** 10.3390/ani14121814

**Published:** 2024-06-18

**Authors:** Giorgio Luis Castellaro, Carla Loreto Orellana, Juan Pablo Escanilla, Nicolás Fuentes-Allende, Benito A. González

**Affiliations:** 1Department of Animal Production, Faculty of Agricultural Sciences, University of Chile, Santiago 8820808, Chile; gicastel@uchile.cl (G.L.C.); carla.orellanam@gmail.com (C.L.O.); juanescanillacruzat@gmail.com (J.P.E.); 2INIA Ururi, Instituto de Investigaciones Agropecuarias, Arica 1001219, Chile; fuente.nicolas@gmail.com; 3Faculty of Forest Sciences and Nature Conservation, University of Chile, Santiago 8820808, Chile

**Keywords:** Andean foothills, cervids, dietary quality, fecal nitrogen

## Abstract

**Simple Summary:**

To develop conservation programs for the taruka (*Hippocamelus antisensis* d’Orbigny), and to evaluate the impacts that this rare species of deer could have in agricultural areas of the Andean foothills of Tarapacá, we studied the composition of its diet, its relative diversity, and its quality in two contrasting seasons, the wet and the dry season. The main components of its diet were alfalfa, followed by herbaceous dicotyledons and shrub species from rangelands, with the presence of grasses and horticultural crop species being very scarce. The above generates a relatively low dietary diversity, with characteristic traits of a concentrate selector herbivore, which could represent a competitive disadvantage for this species. The quality of the diet was relatively stable; however, during the last stage of pregnancy and the beginning of lactation, protein intake, estimated through fecal nitrogen, could limit these processes.

**Abstract:**

The success of conservation programs for the taruka (*Hippocamelus antisensis* d’Orbigny), an endemic and endangered deer, depends on many factors, highlighting anthropogenic and ecological effects. Among the latter, how this herbivore interacts with forage resources is important. The objective of the study was to describe the main attributes of the diet of this deer in rangelands adjacent to agricultural areas of the foothills of the Tarapacá Region, Chile. The botanical composition of the diet (BCD) was determined by microhistology of feces and fecal nitrogen (NF, %) was measured in two contrasting seasons (rainy summer and dry winter). From the BCD and FN, their relative diversity (J) and crude protein percentage were estimated. In the BCD, Medicago sativa dominated (27.6 ± 8.2% vs. 53.9 ± 9.2%, in rainy summer and dry end winter, respectively), followed by herbaceous dicots (46.2 ± 9.4% vs. 19.4 ± 8.7%) and shrubby species (21.5 ± 7.8% vs. 23.4 ± 7.0%), from rangelands. The contribution of grasses and graminoid species was low, not exceeding 3% and 0.4% of the diet, respectively, with no differences between seasons of the year. Intake of horticultural crop species was marginal (1.3 ± 1.3%), being detected only in the wet season. Diet relative diversity was higher during the wet period (0.75 ± 0.07) compared to the dry period (0.58 ± 0.06), since in the first period it was possible to find a greater number of palatable species. There were no significant differences in the FN attributed to the time of the year (average of 1.8 ± 0.19%), which indicates that the diet of this deer would be stable in terms of its protein quality. These FN levels estimate sufficient dietary protein content to satisfy maintenance and early pregnancy, but these could be limiting during late gestation and lactation.

## 1. Introduction

Among the relevant aspects of herbivore ecology, the characteristics of the botanical composition of the diet provide information related to the instantaneous selectivity of forage resources and dietary plasticity in different seasons and ecosystems. Both traits are strongly influenced by climatic variables since these influence plants’ growth and therefore influence the available dry matter to be consumed [1]. This directly affects the foraging behavior of herbivores, who can change their feeding habits, depending on the type of habitat involved [2], as well as the supply of dry matter (DM) of different plants and/or part of them [3].

Concerning the ability of herbivores to modify the components of their diet depending on the temporal abundance of plant species, a distinction can be made between animal species that are efficient in the use of the resource and others [4]. However, the composition of the diet alone does not provide information about its quality, so other indicators have been developed for its evaluation [5]. One of them is the fecal nitrogen content, which has been used as an index of the quality of the diet consumed by herbivores, because it is correlated with dry matter intake, metabolizable energy, crude protein, and even with some parameters associated with rumen activity [6,7,8,9,10,11].

In the present study, we focus on the taruka or northern Andean deer (family Cervidae, subfamily Capreolinae; *Hippocamelus antisensis* d’Orbigny, 1834), one of the least known species of deer in the world. This deer is a medium-sized mammal (around 45 and 80 kg of liveweight in females and males, respectively) endemic to South America that inhabits the gullies and topographically rough areas in the Highland desert and foothill of the Central Andes Ecoregion in Peru, Bolivia, northern Chile, and north-western Argentina [12,13] (Figure 1). Its habitat corresponds to bush steppe environments, with steep rocky slopes and the presence of valleys with water. At an international level, the taruka has been classified as a “Vulnerable” species [14], currently being listed in Appendix I of CITES [15]. The main threats for taruka conservation in Chile are derived from unhealthy interactions with farmers and local communities at the rural areas that they cohabit [16], because almost 90% of the taruka’s habitat is unprotected [17]. Considering that, describing its diet composition and dependency on crops is crucial in contributing to the deer’s conservation in rural areas.

The diet of the taruka has been described qualitatively, particularly in situations of conflict with subsistence agricultural activity, where an unresolved problem is generated [16,18]. It is composed of different species of grasses, leaves, flowers, twigs, succulents, fruits, berries, bark, mosses, lichens, fungi, and seeds [19,20]. Important differences in the taruka’s diet have also been reported between contrasting climatic seasons. In this regard, Gazzolo and Barrio [21], in high-altitude grasslands of the central Andes of Peru, point out that, during the rainy season, grasses were the dominant species of the diet, while, in the dry season, it was dicotyledonous herbaceous species. The absence of quantitative data on plant species consumed and diet quality in Andean foothill environments of the Tarapacá district makes it impossible to confirm their intake strategy (specialist or generalist; grazer or browser), and to measure its impact on the croplands. Therefore, this study analyzes the botanical composition of the taruka’s diet under the hypothesis that the proportions of the different groups of plants consumed change according to dissimilar seasons (rainy summer season versus the dry winter season), which affects the fecal nitrogen content.

This information will allow quantification of its dietary trophic diversity with emphasis on the presence of horticultural and forage crop species in the diet of this cervid for a better understanding of the conflict between the species and local agricultural activity.

## 2. Materials and Methods

### 2.1. Study Area

The study was carried out near Chiapa, a town located within the Volcan Isluga National Park (Tarapacá District, Chile; 19°32′ S; 69°12′ W; 3290 m.a.s.l.), in two contrasting periods: March, representative of the rainy season and October, representative of the dry season. The evaluated area was approximately 40 ha (Figure 2). According to Köeppen’s climate classification (1948) [22], the climate of the sector corresponds to the cold dry desert type (BWk). The average annual precipitation is 169.7 mm, of which 84% occurs between January and March. The average annual temperature is 10.3 °C, with the warmest month being January, with 11.4 °C, and the coldest month being July, with 8.5 °C. The predominant soils of the studied area belong to the order Aridisol (*Typic Haplocambids*), which are characteristic of areas with desert climates where low woody vegetation (xeric shrublands) dominates, with a permanent or almost permanent moisture deficit, added to this is a very low concentration of organic matter, which reflects its low productivity [23]. The predominant natural vegetation corresponds to a low tropical Andean desert scrub, with the presence of some succulents and dominated by low shrubs (<50 cm in height), with little plant cover, belonging to the genera *Atriplex*, *Acantholippia*, and *Ambrosia* [24]. In the areas near the town of Chiapa, the cultivation of different horticultural crop species, cereals, and forage crops (alfalfa or lucerne) is common, which is done on small terraces adjacent to the water courses that exist in the sector [25].

### 2.2. Botanical Composition and Rangeland Cover

In each of the analyzed periods, twenty lines of 50 m length were established to carry out the evaluation of the vegetation using the “Point Transept” method [26]. The lines were marked using a tensioned metric tape, which was placed at an approximate height of 80 cm above the ground surface. In each of the lines, 100 points were evaluated, using a metal needle of 1.3 m in length and 4 mm in diameter, which was dropped perpendicular to the metric tape, to identify in each point the edaphic cover and the presence of each plant species (Figure 3). The relative contribution of each plant species (*CEsp_i_*, %) was calculated using the formula:(1)$${\;CEsp}_{i}=\frac{C_{i}}{\sum_{i=1}^{n}C_{i}}100$$

In (1), *C_i_* is the number of contacts of the n plant species in relation to the total contacts occupied by the plants in each line. With the botanical composition values, the rangeland’s relative diversity was estimated, using Equations (2)–(4), which are described later in the methodology. The calculation of plant cover was carried out from all points that had at least one species. Contact was also recorded with components that did not correspond to points covered by vegetation such as mosses and lichens, bare soil, stones, rocks, and litter. The results for each line were summed and averaged to obtain the botanical composition, relative diversity, and plant cover of the rangeland site.

### 2.3. Collection of Fecal Samples and Determination of the Botanical Composition of the Diet

Ten samples of fecal pellets from tarukas were collected during the dry and wet periods, these samples being approximately 10–15 g. The feces were differentiated in the field from other herbivores (like South American domestic camelids and goats) by their shape and size (Figure 4).

Stool samples were dehydrated in a forced air oven at 70 °C for 48 h and ground at 1 mm in a Willey mill. Subsequently, each sample was split into two portions, one for nitrogen determination and the other to estimate the botanical composition of the diet by microhistological analysis of feces [1,27,28,29]. Five slides per sample were prepared, in which 100 visual fields were evaluated. Fragments of epidermis of plant species were identified from fecal samples using an optical microscope (Olympus, model CX21. Ningbo Huasheng Precision Technology Co. Ltd., Ningbo, China), with a built-in digital camera and magnifications of 100× and 400× [30]. The determination of the fragments of the different plant species was carried out by comparing them with epidermis tissues corresponding to plants belonging to a reference herbarium, prepared with the dominant rangeland’s plant species, horticultural crop species cultivated in the area: garlic (*Allium sativum*), corn (*Zea mayz*), marjoram (*Origanum vulgare*), pumpkin (*Cucurbita moschata*), potato (*Solanum tuberosum*), chard (*Beta vulgaris var. cicla*), cabbage (*Brassica oleracea var. capitata*), beet (*B. vulgaris*), mint (*Mentha piperita*), pea (*Pisum sativum*), wheat (*Triticum aestivum*), celery (*Apium graveolens*), and forage crops such as alfalfa (*Medicago sativa*). The techniques proposed by Castellaro et al. [31] and Catán et al. [32] were used for preparation of reference epidermis samples. The result of the microscopic reading of 100 fields per sample was expressed as relative frequency [27]. The frequency of each species was transformed into density using the Fracker and Brischle tables [27,30]). The identified species were grouped into horticultural crop species, alfalfa, grasses (Poaceae family), graminoids (Cyperaceae and Juncaceae family), dicotyledonous herbs, and shrub species.

### 2.4. Diet Diversity

Diet diversity was obtained using the Shannon–Wiener index:(2)$$H=-\sum_{i=1}^{n}P_{i}\cdot {Log}_{2}(P_{i})$$

In (2), *P_i_* is the proportion of species *i* in the diet and *n* is the number of species in it. The value of *H* was expressed as relative diversity or equality (*J*):(3)$$J=\frac{H}{H_{max}}$$
where *H_max_* is:(4)$$H_{max}={Log}_{2}\left(n\right)\;$$

The variable *H_max_* represents the value that *H* would have if all the species found in the diet had the same frequency [33,34].

### 2.5. Fecal Nitrogen

This analysis was performed on the fraction of the fecal sample destined for this purpose, using the Kjeldahl method [35]. Values were expressed as percentages of organic dry matter basis.

### 2.6. Statistical Analysis

As this study was observational, an “a priori” experimental design was not carried out. However, the proportions of the main groups of species, the relevant species within each group, the value of the dietary diversity index, and fecal nitrogen were analyzed through an analysis of variance, previously checking the assumptions of this, considering the time of year as the only source of variation:(5)$$Y_{ij}=\mu +{Season}_{i}+\varepsilon_{ij}$$

When significant differences existed, the means were separated using Fisher’s Least Significant Difference Test (LSD) with a significance of 5% [36]. Complementary to the above, the nonparametric Spearman correlation matrix was calculated between the analyzed variables to evaluate the degree of association between them.

## 3. Results

### 3.1. Botanical Composition and Range Plant Cover

Both in the rainy and dry seasons, the vegetation cover remained within a very narrow range (25–26%), but with greater variability during the dry period. The three most important dominant species were shrubs of the *Asteraceae* family, especially *Ambrosia artemisioides*. The relative diversity was low, with a range between 31 and 43% in the dry and wet seasons, respectively (Table 1).

The percentage of bare soil ranged between 24.6% in the dry period and 23.7% in the humid period. The presence of stones and rocks was important, totaling 38.6% and 35.1%, respectively. The percentage of mulch was always high, with 13.9% in the dry season and 12.5% in the wet season.

### 3.2. Diet’s Botanical Composition

The diet of the taruka was dominated by the forage species *Medicago sativa* (alfalfa), whose contribution was 27.6 ± 8.2% in the wet season, differing significantly (*p* ≤ 0.001) from the value determined in the dry season, which increased to 53.9 ± 9.2%. In second place were the dicotyledonous herbs, which contributed 46.2 ± 9.4% during the wet season, while, in the dry season, the contribution of this group was significantly lower (*p* ≤ 0.001), averaging 19.4 ± 8.7%. Within this group, species of the genus *Astragalus* were always important, contributing 21.0 ± 8.5% during the wet season, with a lower contribution (15.7 ± 9.2%), although not significant (*p* = 0.1758), during the dry season (Table 2). Shrub species were also important, contributing 21.6 ± 7.8% during the wet season, with a slightly greater contribution, although not significant (*p* = 0.5553) during the dry season, which averaged 23.4% ± 7.0%. Within this group, the contribution of *Baccharis boliviensis*, *Krameria lappacea*, and *Tarasa operculata* was always important. The first species contributed 3.9 ± 2.9% during the wet season, increasing to 8.2 ± 6.2% in the dry season (*p* = 0.0599). In the case of *K. lappacea*, the contribution in the wet season was 3.2 ± 5.0%, increasing significantly (*p* = 0.0167) to 8.7 ± 4.9% during the wet season. In the case of *T. operculata*, its contribution was greater during the wet season with 4.9 ± 4.4%, decreasing significantly (*p* = 0.0086) in the dry season to only 1.1 ± 1.1% (Table 2). The contribution of grasses and graminoid species (Cyperaceae and Juncaceae families) was low, not exceeding 3% and 0.4% of the diet, respectively, with no differences between seasons of the year. Intake of horticultural crop species was marginal (1.3 ± 1.3%), being detected only in the wet season.

### 3.3. Diet’s Diversity

During the wet season, between 13 and 16 species were detected in the diet; this was significantly higher (*p* = 0.0002) than the range obtained in the dry season, which was 10 to 15 species. This number of species together with their respective proportions generated a diversity index (H) of 3.11 ± 0.41 in the wet season, which was significantly higher (*p* ≤ 0.001) than that obtained in the dry season (2.14 ± 0.23). The previous values expressed as relative diversity (J) were 0.746 ± 0.075 and 0.577 ± 0.057, for the wet and dry seasons, respectively, a difference that was significant (*p* ≤ 0.001).

### 3.4. Fecal Nitrogen

The values obtained in the percentage of fecal nitrogen of the taruka (FN, % organic matter base ± standard deviation), were 1.82 ± 0.23 and 1.78 ± 0.17%, for the wet and dry seasons, respectively, there were no significant differences between both evaluation seasons (*p* = 0.6705).

## 4. Discussion

### 4.1. Botanical Composition and Range Plant Cover

The results presented in Table 1 indicate relative stability in the dynamics of vegetation between the evaluated seasons, which maintains contribution values of plant species and cover. The botanical composition is dominated by the shrub component, especially by the species *A. artemisiodes* and *G. tarapacana*, both belonging to the *Asteraceae* family, and according to this, physionomically the vegetation of the evaluated site is a subshrub steppe [37]. The dominant shrub species generally presented little or no consumption by the taruka, which transforms them into species without any nutritional or functional role [38]. The above could be attributed to the presence of resinous and/or aromatic substances, to which is added the leathery structure of their leaves [39], which could induce rejection by animals.

### 4.2. Diet Botanical Composition

The forage crop *Medicago sativa* (alfalfa or lucerne) was the most important dietary item in the two seasons evaluated. This situation can be explained by the scarcity of quality forage resources, especially during the dry season, which forces this herbivore to approach cultivated areas and consume alfalfa sprouts, especially after the winter break. Intake of alfalfa has been reported by Sielfeld et al. [19], where a permanent intake of this forage crop is mentioned throughout the year, but without being an abundant item in the diet. This last aspect differs from what was obtained in this research, where alfalfa intake was always relevant. It is important to highlight the dominance of herbaceous dicotyledons in the diet of the taruka, especially in the wet season. The high contribution of this group of plant species is reported in the diets of deer, such as mule deer (*Odocoileus hemionus hemionus*) and white-tailed deer (*O. virginianus*), which is explained by the low capacity that these cervids have for digesting fibrous elements [1], an aspect that could also be valid for the taruka.

Within this group of plants, the high percentage of *Astralagus* sp. stood out during both periods evaluated; it could be related to the scarcity of food resources, which forces the intake of species with low palatability and nutritional value; however, the intake of this legume could be conditioned by the presence of alfalfa, as a negative and significant correlation was detected (r = −0.540; *p* = 0.013) in the dietary contributions of both species. It is important to note that *Astragalus* sp. would be selected by this cervid, considering its high dietary contribution compared to its presence in the rangeland, so it would contribute to improving the quality of the taruka’s diet. However, it should be mentioned that many species belonging to the genus *Astragalus* have secondary metabolites in their tissues [40,41], which have been reported as toxic in domestic livestock [42]. This situation suggests that taruka could have a greater capacity to detoxify substances such as those mentioned compared to domestic ruminants. Shrubs were important in the diet of the taruka, with a relatively similar contribution between the two seasons of the year evaluated. Holechek et al. [1] indicate that browsing deer preferentially consume grasses and shrubs, regardless of the locality in which they are found, which partly coincides with our results. In this regard, the shrub *B. boliviensis*, with a high contribution to the vegetation (see Table 1), would be a species selected in the dry season, but during the wet season it would tend to be rejected. *G. tarapacana* and *A. artemisioides*, the dominant shrubs in the rangeland, are probably species with negative selectivity, since they were not detected in the diet, while *K. lappacea* would be selected in both seasons. Ruminants classified as “concentrate selectors or browsers” [43] can present digestive disorders if they are forced to consume large quantities of grasses in advanced stages of maturity (with high fiber content), which is why they consume limited quantities of grasses during the spring when the herbaceous dicotyledonous plants and shrubs are scarcely available. In the present study, and in both evaluation periods, very low amounts of grasses were detected in the diet, which reaffirms what was previously stated. Browsing herbivores consume plant tissues with a high concentration of cellular contents and that are low in fiber, given that their stomach anatomy is adapted to this situation [44,45], which could be the case of the taruka.

The botanical composition of the taruka diet in the Andean foothills environment differs from that reported by Gazzolo and Barrio [21], who studied the taruka diet of Huascarán National Park, Central Peru. These authors mention that the diet of these cervids was made up of more than 50 species and was dominated by herbaceous dicotyledons in the dry season and by grasses during the rainy season, highlighting in both seasons the species *Werneria nubigena*, *Poa gymnantha*, *Senecio comosus*, and *Ephedra americana*, reflecting a concentrate selector behavior at least during the dry season. These differences could be attributed mainly to the dissimilarity in the habitat between both sectors, since, in the study carried out in Peru, the dominant vegetation is bunch perennial grasses, unlike what happened in this study, where the subshrub steppe was the dominant plant formation, with very little presence of grasses and herbaceous dicotyledons (Table 1), typical of a relatively more arid environment compared to that of the aforementioned study. When comparing the diet of taruka reported in this research with that of huemul, a species of deer of the same genus (*H. bisulcus*) and which was determined to be in the Nevados de Chillán-Laguna Laja biological corridor [46], there is a greater correspondence, since in these last cervids, levels of shrub and tree species in the diet are reported in the order of 15.8% and 13.8%, respectively, while the herbaceous species contribute 70.4%, highlighting within this last group legume species of the genus *Adesmia*. High levels of shrub (58.6%) and tree species (19.5%) are reported in the huemul’s diet from the Nahuel Huapi National Park (Argentina), with the contribution of herbs and grasses being around 9% and that of graminoids around 1% [47]. Vila et al. [48], on huemul’s diet, also report an important contribution of tree and shrub species, but with a low contribution of dicotyledonous herbs and grasses.

### 4.3. Diet’s Diversity

The higher values of the dietary diversity index (J) obtained during the rainy period could be explained by the possibility of finding a greater supply of palatable species in the rangeland during this period, especially dicotyledonous herbs, which would be reflected in the diet. However, these figures are lower than those found in other highland ungulates, such as vicuñas (*Vicugna vicugna* Mol.) and guanacos (*Lama guanicoe* Müller), where J reaches values of 0.8 [49,50]. This aspect would indicate that the taruka therefore present a lower dietary plasticity, a narrower trophic niche, and a lower capacity for competition compared to domestic camelids, which would become more acute in the dry period. Similar conclusions are reported by Vila et al. [51] for the case of huemuls interacting with domestic livestock (cattle and sheep) in grazing lands of Los Alerces National Park, Argentina. Despite what was stated above, it should be kept in mind that a high intake of alfalfa could condition the previous results, since there was a negative and significant correlation (r = −0.892; *p* ≤ 0.001) between this index and the percentage contribution of this foraging to the taruka diet.

### 4.4. Fecal Nitrogen

The lack of differences between the fecal nitrogen (FN) obtained between both seasons would indicate that the taruka diet would be relatively stable from the point of view of its protein quality. The average levels of this indicator were of the order of 1.80 ± 0.19%, lower values than those reported by Osborn and Ginnett [7], in the case of white-tailed deer, and those reported by Kie and Burton [52], in black-tailed deer (*O. hemionus columbianus*). Kamler and Homolka [8] report FN values of around 1.54% for the feces of the European roe deer (*Capreolus capreolus*) during the winter season. The variability obtained in the fecal N content in our study (~11%) could be explained by the influence of, in addition to nutritional variables, factors such as sex, age, and stage of gestation [53], aspects that we could not be cover in our study. Such factors would be important to measure in future research.

By using the average FN value obtained in this study and the equations cited by Hodgman et al. [54], for mule deer (PC_d_ = 3.253 + 13.346·Ln(FN), we estimate a dietary crude protein content (PCd, %) of the order of 11.1%. This value would meet the maintenance requirements of adult and juvenile deer, as well as those required by adult females during early pregnancy, but would be insufficient to meet the requirements of fawns with high growth rates, those of lactating females, as well as those of adult males during the period of antler development [55,56].

In addition to these results, it should be noted that there was no species identified in the diet that was significantly correlated with FN values.

## 5. Conclusions

Our preliminary results indicate that there are differences in the percentage of the different groups of plant species that make up the diet of tarukas depending on the year season, with alfalfa being the most important species, while the dietary contribution of horticultural crops would be marginal. Therefore, the conflict between the taruka populations and local agriculture would be related to alfalfa crop. Within the rangeland species, the dietary contribution of herbaceous dicotyledons and shrub species is relevant, so the taruka could be classified as a “concentrate selectors or browser-type” ruminant. This behavior could be conditioned by the consumption of alfalfa, which can negatively affect the diversity of its diet, which is lower compared to other sympatric ungulates in arid ecosystems of the Andean foothills and highlands. Due to the above, the taruka could be more sensitive to competition with these, especially with domestic camelids, which would become more acute during times when forage resources are scarcer. Apparently and according to the FN levels quantified in this study, the protein quality of the taruka diet does not vary substantially between seasons of the year and would contain sufficient N levels to satisfy maintenance and early pregnancy. These levels could be limiting during late pregnancy and early lactation. The results of this research, being preliminary in nature, should be ratified in future studies using a greater number of observations and comparing the diet of the taruka with other sympatric herbivores.

## Figures and Tables

**Figure 1 animals-14-01814-f001:**
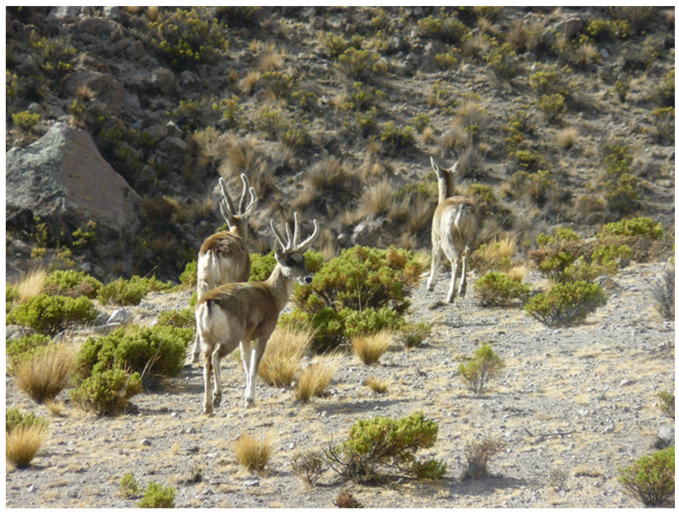
Tarukas (*Hippocamelus antisensis*) in their natural habitat, in rangelands of the Andean foothill of northern Chile (photo by G. Castellaro).

**Figure 2 animals-14-01814-f002:**
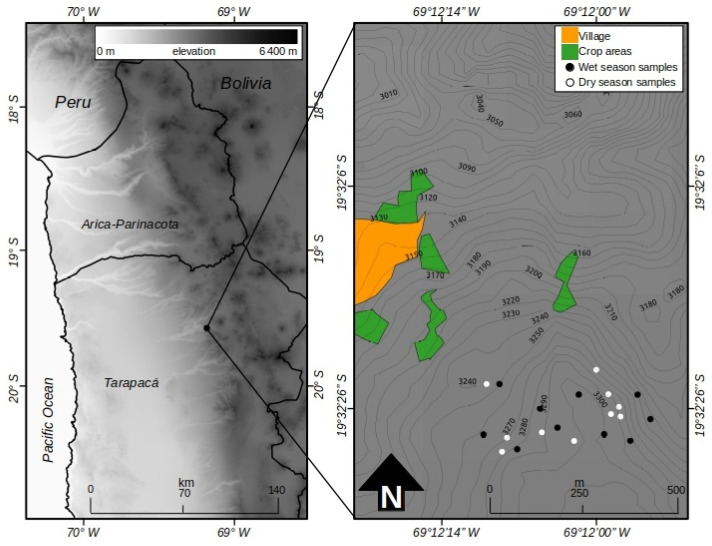
Location of area where the study was carried out.

**Figure 3 animals-14-01814-f003:**
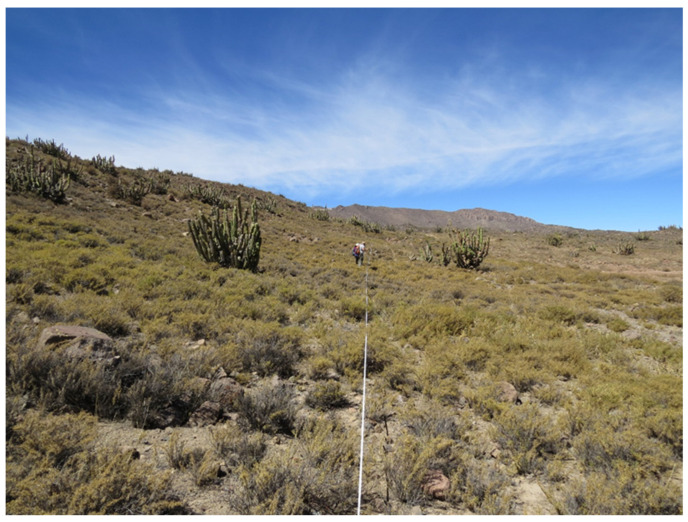
Linear transept used to evaluate plant cover and range botanical composition (photo by J. P. Escanilla).

**Figure 4 animals-14-01814-f004:**
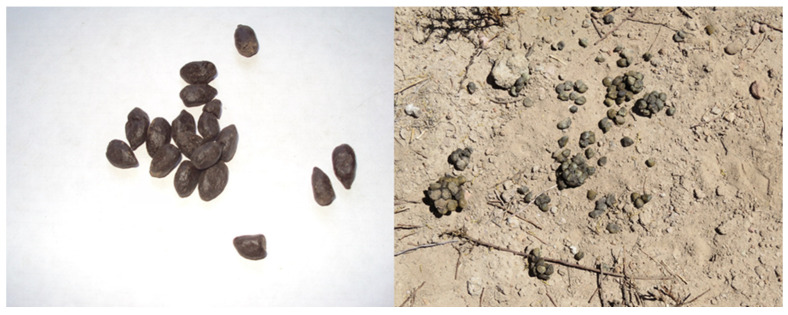
Taruka feces where its characteristic shape stands out (photo by C. Orellana).

**Table 1 animals-14-01814-t001:** Botanical composition and vegetation cover (average ± standard deviation) of the rangeland of the study area in two evaluation seasons. Chiapa, Tarapacá Region, Chile.

Vegetal Species	Season	
	Wet Season (%)	Dry Season (%)
*Ambrosia artemisiodes*	64.0 ± 46.0	75.4 ± 38.5
*Grindelia tarapacana*	17.8 ± 28.1	19.5 ± 37.5
*Baccharis boliviensis*	12.2 ± 19.6	1.0 ± 2.6
*Astragalus* sp.	0.8 ± 2.0	0.0
*Balbisia microphylla*	0.7 ± 1.8	2.8 ± 6.8
*Cumulopuntia sphaerica*	0.7 ± 1.6	0.0
*Fabiana ramulosa*	1.7 ± 4.1	0.0
*Stipa* sp.	0.8 ± 2.0	0.0
*Tagetes multiflora*	1.3 ± 3.1	1.3 ± 2.3
Total	100.0	100.0
Plant cover	25.3 ± 6.5	26.3 ± 10.2
Bare soil	23.7± 5.9	24.6 ± 9.7
Stones	10.2 ± 6.9	11.8 ± 14.8
Rocks	28.4 ± 8.0	23.3 ± 8.7
Litter	12.5 ± 7.3	13.9 ± 9.1
Relative diversity (%)	43.0 ± 28.4	31.2 ± 32.2

**Table 2 animals-14-01814-t002:** Diet’s botanical composition of taruka (*Hippocamelus antisensis*) during the wet and dry season. Chiapa, Tarapacá Region, Chile.

Plant Species	Wet Season	Dry Season	*p*-Value ^1^
	Mean ± SD	Mean ± SD	
Forage species			
*Medicago sativa*	27.6 ± 8.2	53.9 ± 9.2	<0.001
Horticultural crop species	1.3 ± 1.3	0.0 ± 0.0	
*Allium sativum*	0.1 ± 0.2	0.0 ± 0.0	
*Zea mays*	0.3 ± 0.9	0.0 ± 0.0	
*Origanum vulgare*	0.9 ± 1.1	0.0 ± 0.0	
*Cucurbita moschata*	0.1 ± 0.4	0.0 ± 0.0	
Grasses	2.5 ± 2.2	3.2 ± 1.5	0.4317
*Cortaderia jubata*	0.0 ± 0.0	0.0 ± 0.0	
*Bromus catharticus*	0.1 ± 0.2	0.0 ± 0.0	
*Stipa* sp.	0.7 ± 0.8	0.1 ± 0.1	
*Polypogon australis*	0.7 ± 1.0	3.1 ± 1.4	
*Poa* sp.	0.0 ± 0.0	0.0 ± 0.0	
*Stipa leptostachya*	0.2 ± 0.8	0.0 ± 0.0	
*Muhlenbergia asperifolia*	0.2 ± 0.8	0.0 ± 0.0	
*Stipa mucronata*	0.6 ± 1.1	0.0 ± 0.1	
Graminoids	0.7 ± 1.0	0.1 ± 0.1	0.0453
*Isolepis cernua*	0.0 ± 0.1	0.0 ± 0.0	
*Scirpus asper*	0.7 ± 1.0	0.1 ± 0.1	
Dicotyledonous herbs	46.3 ± 9.5	19.4 ± 8.7	<0.001
*Convolvulus arvensis*	1.9 ± 1.7	0.0 ± 0.0	
*Schinus molle*	0.1 ± 0.2	0.0 ± 0.0	
*Erodium* sp.	5.3 ± 6.8	0.0 ± 0.0	
*Tagetes multiflora*	1.5 ± 3.0	0.2 ± 0.4	
*Astragalus* sp.	21.0 ± 8.5	15.7 ± 9.2	0.1758
*Pellaea ternifolia*	6.6 ± 8.3	0.1 ± 0.2	
*Bidens andicola*	0.2 ± 0.4	0.2 ± 0.4	
*Lupinus tarapacensis*	0.2 ± 0.3	1.1 ± 0.8	
*Otholobium pubescens*	0.4 ± 1.1	0.0 ± 0.0	
*Spergularia fasciculata*	1.3 ± 0.9	0.0 ± 0.0	
*Aldama helianthoides*	2.3 ± 4.8	0.3 ± 0.4	
*Malva parviflora*	2.0 ± 3.0	0.0 ± 0.0	
*Aldama* sp.	1.3 ± 2.0	0.0 ± 0.0	
*Marrubium vulgare*	0.0 ± 0.0	0.0 ± 0.0	
*Althaea rosea*	2.3 ± 2.3	0.0 ± 0.1	
*Matricaria recutita*	0.1 ± 0.2	0.0 ± 0.0	
*Montiopsis* sp.	0.0 ± 0.0	1.8 ± 2.1	
Shrubs	21.5 ± 7.8	23.4 ± 7.0	0.5553
*Atriplex glaucescens*	0.2 ± 0.8	0.0 ± 0.0	
*Balbisia microphylla*	0.2 ± 0.4	0.6 ± 0.8	
*Fabiana ramulosa*	0.1 ± 0.3	0.0 ± 0.1	
*Baccharis boliviensis*	3.9 ± 2.9	8.2 ± 6.2	0.0599
*Grindelia tarapacana*	1.1 ± 1.3	0.1 ± 0.2	
*Senna birostris var. Arequipensis*	0.7 ± 1.6	0.1 ± 0.3	
*Ambrosia artemisiodes*	1.1 ± 1.6	0.1 ± 0.4	
*Tarasa operculata*	4.9 ± 4.4	1.1 ± 1.1	0.0086
*Senecio coscayamus*	3.0 ± 2.0	0.4 ± 0.6	
*Krameria lappacea*	3.2 ± 5.0	8.7 ± 4.9	0.0167
*Escallonia angustifolia*	0.1 ± 0.3	0.2 ± 0.4	
*Junellia arequipensis*	0.7 ± 1.0	0.0 ± 0.0	
*Aphyllocladus denticulatus*	0.0 ± 0.1	0.0 ± 0.0	
*Baccharis juncea*	1.4 ± 4.5	0.0 ± 0.0	
*Ephedra breana*	0.8 ± 2.4	3.8 ± 3.5	

^1^ The analysis of variance was carried out at the level of groups of species and, within each group, at the most important species.

## Data Availability

Original data can be requested directly from the authors.

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
