# Peer review of "Preliminary Report on Diet Estimation of Taruka (Hippocamelus antisensis d’Orbigny, 1834) in an Agricultural Area of the Andean Foothills of the Tarapacá Region, Chile"

_animals, 2024, doi:10.3390/ani14121814_

Round 1

Reviewer 1 Report

Comments and Suggestions for Authors

The ms is worth publishing after minor changes. The authors should better emphasize the peculiarities of their results. Taruka appears here to be a browser, a concentrate selector. Is this exactly the same in other areas of study? Is taruka here more selective due to the particularly arid climate which obliges it to carefully choose the plants with more nutrients? Or is this a general rule for all the studied habitats?

I would add some notes and corrections on specific points:

Line 2 d’Orbigny. The correct writing (see also lines 22, 61, Fig.1 etc) is with a lowercase letter for d and a capital letter for O. It is not necessary to repeat everywhere the name of the French zoologist who named the species.

Lines 19-21: Please reformulate the whole sentence in a better and clearer English

Line 60: taruka or northern Andean deer

Line 62: around 45 and 80 kg of mean liveweight

Line 113: alfalfa or lucerne

283: their leaves

286: diet botanical composition

297-98: Please correct the English

305: “this species…by this cervid” please reformulate

320: please cancel “species”

326: please eliminate “in which this study was carried out”

346: …Vila et al. [47] on huemul’s diet

355: The taruka therefore presents a lower dietary plasticity…

388: selector

396: “said levels” No, please reformulate

388-89: Please develop your discussion: Is taruka a concentrate selector also in other study areas? More in your area than in others? Please explain why?

Line 434: I would suggest you to add also Barrio J 2010 Taruka Hippocamelus antisensis (d’Orbigny 1834) pages 77-88 in Duarte JMB and González S. eds. Neotropical Cervidology. FUNEP

Comments on the Quality of English Language

The English is not particularly fluent and concise and should be slightly improved. 

Author Response

Thank you for all your suggestions regarding our manuscript “Diet of taruka (Hippocamelus antisensis d'Orbigny, 1834) in agricultural areas of the Andean foothills of the Tarapacá Region, Chile”, by the authors Giorgio Castellaro, Carla Orellana, Juan Escanilla, Nicolás Fuentes -Allende, Benito A. González. We have reviewed each of their observations, which we incorporated into the new manuscript highlighted with red letter. The answers to each of them are detailed in pdf archive.

Reviewer 2 Report

Comments and Suggestions for Authors

This is an interesting study but needs a few improvements.

L22-24. Reformulate this sentence. It seems that the main conservation threat for the species is connected to feeding resources, which is incorrect as far as I know about the species. Several other factors are the same or probably more important.

L45. It was surprising for me that the authors introduced the term “selectivity of forage resources” at the beginning of the manuscript, considering that there are no selectivity analyses. The authors have data about the consumption and the plant availability (line transects). With this data, it is possible to calculate some selectivity indexes like Ivlev or Manly. These would make the results more interesting since the reader can know not just what taruka is eating in the area but also what they prefer (or reject) to eat.

L48. “availability” + “available”.

L56-59. This is a highly repeated sentence in such kind of studies still far from evident. The faecal N is not just derived from the dietary N, but also from the fibre digestibility leading to an increase in the microbial N (just as a very basic initial fact). I would recommend reading several papers on this topic by Marcus Clauss and his team or the recent papers by Stipan ÄŒupic identifying the many external factors affecting faecal N.

L70. “they”

L83-87. In line with the previous comment, this hypothesis is very unclear. Does “without variation between seasons” refer to the seasonal diet or the seasonal faecal N? Is it really expectable that the seasonal diet changes correlate with the faecal N? The contrary is the logic to me: the animals change their diet seasonally in order to select the plants necessary to help them meet their N requirements (further comments about this topic later), so I would expect changes in the diet without changes in the fN.

L102-105. Please provide the range of years from which these average values were obtained.

L113. Alfalfa is here defined as “forage crop”, which is correct. Then, it is treated as forage in Table 2 (separated from crops), concluding that (L247-248) “the intake of crop species is marginal”. A reader can extract from this that there is no conflict with the local farmers, which seems incorrect to me. Is the alfalfa growing naturally in the area? Or is it grown by farmers to feed their livestock? I guess the second is right. Thus, it is a crop and there is conflict. “Forage crop” because is it used as forage for livestock, but still crop. This should corrected everywhere in the manuscript not to lead to wrong conclusions.

L126-130. Please, indicate the height considered for this method. It should count all the biomass available for the studied species, according to their size.

L149-152. If the samples could have been confounded with other ungulates, please, provide a list of these other species in the area.

L203-206. Please indicate the exact test which was conducted. Later on, there is indication about Spearman correlation, so I guess it was some non-parametric test.

L229. The singular of species is species, not specie. “Spice” means something totally different.

L256-258. 13-16 species doesn´t look significantly lower than 10-15…

Figure 7. In line with previous comments, the authors interpret this figure as “no seasonal fN differences”. For me, there is something even more important here, which is greater fN variability in the wet season. I previously recommended the papers by Stipan ÄŒupic. These can help to explain this result. It would be interesting for me, as a reader, to understand how this can be influenced by the yearly biology of the species. Is it a period with greater requirements by females (late gestation, lactation)? That would explain why part of the population selects for a higher diet quality.

Author Response

Thank you for all your suggestions regarding our manuscript “Diet of taruka (Hippocamelus antisensis d'Orbigny, 1834) in agricultural areas of the Andean foothills of the Tarapacá Region, Chile”, by the authors Giorgio Castellaro, Carla Orellana, Juan Escanilla, Nicolás Fuentes -Allende, Benito A. González. We have reviewed each of their observations, which we incorporated into the new manuscript highlighted with green letter. The answers to each of them are detailed in pdf archive.

Reviewer 3 Report

Comments and Suggestions for Authors

The study is interesting, but 10 samples are insufficient to obtain reliable results, make comparisons and build diet indexes. The information provided by only 10 samples is too limited and cannot be considered representative of the species' diet in any case. You need more samples and it would be interesting to compare the diet not only betwen sexes but also depending on the age, for example.

Comments on the Quality of English Language

The article is correctly written and provides sufficient information.

Author Response

Thank you for all your suggestions regarding our manuscript “Diet of taruka (Hippocamelus antisensis d'Orbigny, 1834) in agricultural areas of the Andean foothills of the Tarapacá Region, Chile”, by the authors Giorgio Castellaro, Carla Orellana, Juan Escanilla, Nicolás Fuentes -Allende, Benito A. González. The response to your comment is detailed in pdf archive.

Round 2

Reviewer 2 Report

Comments and Suggestions for Authors

I´m satisfied with the new draft.

Author Response

Dear Revisor2

Thank you very much for your relevant contributions which helped improve our manuscript.

 best regards

Reviewer 3 Report

Comments and Suggestions for Authors

I still believe that 10 samples are not sufficient for crafting a scientific article, let alone for extracting indices like the Shannon index. 10 samples are neither adequate nor representative for addressing the diet of a species. That sample size is not sufficient to draw conclusions from the article. For example, when you state, 'The lack of differences between the Fecal nitrogen (FN) obtained between both seasons would indicate that the taruka diet would be relatively stable from the point of view of its protein quality,' you do not have enough data to make this assertion. With only 10 samples, nothing can be categorically affirmed, and it is very difficult to detect differences between effects due to individual variability and variations among sexes, ages, etc. This is why a large sample size is needed. 

Perhaps it could be a short note or something similar, but not a full research article. I understand that it concerns a threatened species, but we are not discussing a population of 100 individuals; there are at least 1500 remaining in the wild.

Comments on the Quality of English Language

The article is well-written.

Author Response

Response to Revisor 3.

Once again, we appreciate your suggestions and comments. We are aware that the number of observations we used in our research was limited, but it is what we were able to find in the area where this work was carried out. Due to the above, the results of our research will be considered preliminary. Due to the above, and at the suggestion of the Academic Editor of the Journal Animals, our article will be presented as a Brief Report. In this regard, in our conclusions we make reference to this situation, recommending that the number of observations be increased in future studies.

However, we believe that our work is a contribution since it at least accounts for the magnitude of the variability that the relevant characteristics of the taruca diet present, which will allow future work to better dimension the required sample sizes.

Best regard
